# Aptamers Increase Biocompatibility and Reduce the Toxicity of Magnetic Nanoparticles Used in Biomedicine

**DOI:** 10.3390/biomedicines8030059

**Published:** 2020-03-14

**Authors:** Galina S. Zamay, Tatiana N. Zamay, Kirill A. Lukyanenko, Anna S. Kichkailo

**Affiliations:** 1Federal Research Center “Krasnoyarsk Science Center of the Siberian Branch of the Russian Academy of Science”, 660036 Krasnoyarsk, Russia; kirill.lukyanenko@gmail.com; 2Laboratory for Biomolecular and Medical Technologies, Krasnoyarsk State Medical University named after prof. V.F. Voino-Yasenecki, 660022 Krasnoyarsk, Russia; tzamay@yandex.ru

**Keywords:** aptamers, magnetic nanoparticles, biocompatibility, toxicity

## Abstract

Aptamer-based approaches are very promising tools in nanomedicine. These small single-stranded DNA or RNA molecules are often used for the effective delivery and increasing biocompatibility of various therapeutic agents. Recently, magnetic nanoparticles (MNPs) have begun to be successfully applied in various fields of biomedicine. The use of MNPs is limited by their potential toxicity, which depends on their biocompatibility. The functionalization of MNPs by ligands increases biocompatibility by changing the charge and shape of MNPs, preventing opsonization, increasing the circulation time of MNPs in the blood, thus shielding iron ions and leading to the accumulation of MNPs only in the necessary organs. Among various ligands, aptamers, which are synthetic analogs of antibodies, turned out to be the most promising for the functionalization of MNPs. This review describes the factors that determine MNPs’ biocompatibility and affect their circulation time in the bloodstream, biodistribution in organs and tissues, and biodegradation. The work also covers the role of the aptamers in increasing MNPs’ biocompatibility and reducing toxicity.

## 1. Introduction

Recently, magnetic nanoparticles (MNPs) have begun to be successfully applied in various fields of biomedicine. They are used for MRI imaging of pathological foci, as an iron supplement for the treatment of iron deficiency anemia [1], in the labeling and retention of mesenchymal stem cells at implantation site or to engineer organized tissues [2,3,4], the targeted delivery of therapeutic agents, magnetomechanical stimulation of bone tissue regeneration, skin regeneration [5], etc. The application of technologies based on biomagnetic nanoparticles has allowed the development of methods for the differentiation of human osteoblasts with an external alternating magnetic field. Thus, MNPs are becoming promising tools for a wide range of applications in biomedicine and, in particular, regenerative biomedicine [6,7,8,9,10].

It should be noted that MNPs are natural components of living systems; they are synthesized in the cells of bacteria, fish, birds, and humans. In humans, MNPs have been found in various types of cells—the brain, heart, spleen, liver, bone tissue and tumors [11,12,13], nevertheless, their exact role, as well as the cause of their occurrence, are not fully understood [14]. It can be assumed that endogenous magnetic nanoparticles are a cellular depot of iron and, in addition, are involved in the differentiation of cells by changing its membrane potential.

The use of exogenous biomagnetic nanoparticles is limited by their potential toxicity [15], which is due to their biocompatibility. This biocompatibility depends on many factors: chemical nature, coating, biodegradability, the compatibility of surfactants of magnetic nanoparticles with the environment [16], solubility, pharmacokinetics, targeted delivery mechanism, the chemistry of surface phenomena, structure, stability of colloidal solutions of nanoparticles, amount of injected nanoparticles and their ability to integrate into the patient’s body without causing adverse clinical manifestations and inducing a cellular or tissue response necessary to achieve optimal therapeutic effect and shape [17,18] (Figure 1). In the absence of biocompatibility, nanoparticles disrupt cellular and tissue metabolism, causing toxic effects. Therefore, the biocompatibility of MNPs should be considered as the primary requirement for their use in biomedicine.

Currently, the most popular studies of MNPs’ biocompatibility are in vitro studies, although MNPs may behave differently under the conditions of the body [19], since the blood, where biomagnetic NPs are injected, is a highly ionized heterogeneous medium [20]. Therefore, when MNPs enter the bloodstream they can (1) stick together; (2) change their magnetic properties; (3) react with plasma proteins, intercellular substances and cells that are not the target of their delivery.

Opsonization is one of the main factors determining the MNPs’ biocompatibility and their circulation time in blood plasma. The level of opsonization of MNPs depends on their: (1) hydrophobicity [21]; (2) charge; (3) size and shape [22]. The opsonized MNPs are removed from the bloodstream within a few minutes. Among them, 80%–90% enter the liver, 5%–8%—the spleen, 1%–2%—the bone marrow [21]. A decrease in the opsonization of MNPs is achieved by increasing the hydrophilicity of nanoparticles. Hydrophilic and neutral MNPs are not recognized by macrophages and have a longer plasma circulation time.

The next limitation in the use of MNPs at the organism level is the presence of barriers during their transition from blood vessels to the lymphatic system and tissues [23], as well as when entering the cells [24]. The type, structure, and geometry of the MNPs determines their effectiveness in overcoming these barriers [25]. The hydrodynamic dimensions of MNPs affect their distribution inside the blood vessels, the mechanism for removing particles from the body, and ways to overcome biological barriers. A decrease in the size of the spherical particles leads to an increase in their concentration in the center of the blood vessels and a decrease in their interaction with the walls of the vessels. This results in an increase in the circulation time of MNPs in the blood [26]. The hydrodynamic size also affects the removal of particles from the body [20]: small particles (less than 20 nm) are excreted by the kidneys, medium-sized particles (30–150 nm) accumulate in the bone marrow, heart, kidneys and stomach, large particles (150–300 nm) accumulate in the liver and spleen. The influence of the nanoparticles’ shape on their circulation time in the body is presented in a number of studies [27].

Iron-based nanoparticles are the most biocompatible [28], since iron is easily degraded in the body. In addition, the methods of introducing MNPs into the body affect their toxicity and biocompatibility. In particular, the intranasal administration of MNPs causes morphological changes, the activation and proliferation of microglia in the olfactory bulb, hippocampus and striatum [29], and the acute oral administration of MNPs leads to the inhibition of acetylcholinesterase, Na^+^-K^+^-pump, and also Mg^2+^- and Ca^2+^-ATPase in rats [30]. The coating of MNPs is an essential factor determining their biocompatibility [31].

## 2. The Mechanisms of the Toxic Effect of MNPs

MNPs can contribute to the development of toxic effects such as oxidative stress, embryotoxicity, mutagenicity, genotoxicity and vascular embolism. This is due to MNPs’ agglomeration in the bloodstream, the activation of the immune system, inflammatory processes [32], changes in cell morphology, impaired cell signaling and stem cell differentiation, and damage to the cytoskeleton [33] (Figure 2). The mechanisms of the toxic effect of MNPs are not fully understood.

Oxidative stress is one of the most likely toxicity mechanisms [34,35,36]. The reason for its occurrence is the formation of reactive oxygen species (ROS) in the Fenton reaction Fe^2+^ + H_2_O_2_ = Fe^3+^ + OH^−^ + OH^−^ in the interaction of magnetite with cells [17,37]. Iron ions are formed in the cell during the decomposition of magnetite in lysosomes, and the resulting hydroxyl radicals can enter into undesirable reactions with DNA, proteins, polysaccharides and lipids.

In addition, ROS can be generated from the surface of MNPs by the leaching of metal ions or the release of oxidizing agents by the enzymatic degradation of MNPs. The accumulation of ROS destroys cellular proteins, enzymes, lipids and nucleic acids and, as a result, contributes to the disruption of cellular processes, leading to apoptosis and necrosis [15]. ROS can disrupt mitochondrial activity, alter membrane potential and cause morphological changes in the mitochondria, which can have an adverse effect on cell viability, proliferation and metabolic activity [37]. Reactive oxygen species such as superoxide-anion O2^٠−^, hydroxyl radical and singlet oxygen, lead to DNA damage when interacting with it [38].

In vitro studies have established that nanoparticles with strong oxidizing (CeO_2_, Mn_3_O_4_, Co_3_O_4_) or reducing (Fe_0_, Fe_3_O_4_, Ag_0_, Cu_0_) properties can be cytotoxic and genotoxic with respect to their biological targets. One of the main sources of toxicity is electronic and/or ionic transfer, which occurs during oxidation-reduction, dissolution, and catalytic reactions either inside the crystal lattice of nanoparticles or when it enters the culture fluid [39].

MNPs can alter cellular metabolism, in particular, increasing the secretion of IL-1β, IL-6, and TNF-α in rat primary microglia [40], decreasing the secretion of IL-1β, but not TNF-α, in primary mouse microglia, causing a short-term increase in NO production [29]. The neurotoxicity of MNPs needs to be considered. The data on the effect of MNPs on neurons are controversial [41,42,43]. Iron oxide MNPs affect neuron function in a dose-dependent manner, enhancing PC12 cells outgrowth [41] or diminishing the viability and capacity of these cells [43]. High concentrations of MNPs disrupt the glucose metabolism, and alter the expression of genes associated with the metabolism of organic acids (OAs) and amino acids (AA) in HEK293 cells [44].

MNPs can damage the cytoskeleton, and disrupt the integrity of cell membranes or native transport chains of ions and electrons [39]. As a result, multiple side effects in cells exposed to superparamagnetic iron oxide nanoparticles may occur, including mitochondrial function weakening, the activation of inflammatory processes, stimulation of apoptosis, biologically active molecules leakage through the cell membrane, the activation of reactive oxygen species generation, and the increase in micronuclei number (which indicates damage to chromosomes and genotoxicity), as well as the condensation of chromosomes [17].

## 3. Aptamers Increase the Biocompatibility of Magnetic Nanoparticles

One of the most important factors that increases the biocompatibility of magnetic nanoparticles is their functionalization, since it: (1) prevents the opsonization of magnetic particles; (2) changes their charge and shape; (3) shields Fe ions that cause oxidative stress; (4) reduces immunotoxicity; (5) leads to the accumulation of MNPs only in the necessary organs, providing targeted binding of magnetic particles only to molecular targets; (6) reduces the risk of thrombosis due to the prevention of the agglomeration of MNPs in the bloodstream.

Peptides, aptamers, antibodies, polysaccharides and small molecules in the form of acids are used as target ligands with functional properties. Ligands provide magnetic nanoparticles with new properties, increase their circulation time in the bloodstream, and facilitate their targeted delivery to the pathological focus.

Peptides are short amino acid sequences that are affine to cellular receptors or components of the extracellular matrix of tissues [45]. Antibodies with high specificity for their targets and great diversity are widely used to ensure the targeted delivery of MNPs to target cells [46]. However, particles conjugated to antibodies have two main disadvantages: (1) the large size of the antibodies (about 20 nm), which reduces the diffusion rate through biological barriers and (2) the immunogenicity of antibodies, which can cause an immune response in the body.

Aptamers are the most promising ligands for the functionalization of MNPs and increase in their biocompatibility. Aptamers are synthetic single-stranded RNA or DNA molecules (30–80 nucleotides in size) capable of specific binding to any molecular and cellular targets: proteins, small organic molecules, viral particles, bacteria, antibodies, whole cells, cell lysates and even tissues [47,48,49,50,51]. They are made using SELEX technology, which allows the targeted selection of oligonucleotides that have an affinity for a given biological target. Aptamers are functional analogs of antibodies. However, due to their physicochemical properties and the preparation method, they have several advantages over antibodies, such as a high specificity, stability, low immunogenicity, ability to withstand reversible denaturation and low production cost. In addition, aptamers bind to large and small targets, while antibodies mainly bind only to large molecules. Aptamers improve the biocompatibility of MNPs by increasing their solubility and preventing agglomeration [52].

The biocompatibility criterion for the studied materials, including MNPs, is considered to have a survival rate of more than 80% of cells after their exposure to the material. In the study of biocompatibility, this criterion was used to assess the toxicity of magnetic nanoparticles [53]. One of the most important factors in the toxicity of MNPs is their concentration [8,17,54,55,56,57,58]. It was shown that magnetite nanoparticles at a concentration of 50 μg/mL, both uncoated and coated with dextran, caused cell death and reduced the proliferation of fibroblasts [59]. While MNPs @ SiO_2_ with the same size, but in lower concentrations, did not cause toxicity in vivo, they penetrated through the blood–brain barrier (BBB), were found in all tissues and remained in the body for a long time [60]. The same data on the toxicity threshold of MNPs were obtained in other studies, which showed that MNPs used at a concentration of 60 μg/mL exhibited cytotoxicity and a reduced cell survival rate [61,62].

Aptamers significantly increase the toxicity threshold. It was shown that magnetic nanoparticles functionalized by aptamers, even at high concentrations (100–200 μg/mL), did not exhibit cytotoxicity in vitro and in vivo [52,63,64]. Moreover, aptamers assisted in the cytotoxicity reduction in PbS-based nanoparticles. In particular, it was shown that nanoparticles functionalized by aptamers did not change cell viability in vitro, while non-functionalized nanoparticles based on lead acetate showed a pronounced cytotoxic effect [65].

Targeted exposure to the pathological focus is the main requirement of personalized biomedicine. The biocompatibility and toxicity of any drug depends on its effect on healthy tissues, and, therefore, on its distribution and accumulation in organs and tissues. Aptamers affined to their molecular target promote the active transfer of MNPs only to target organs [63,65]. This strategy of MNP transfer, in comparison with passive transfer, provides a high specificity and efficiency of MNP delivery to target cells, avoiding non-specific binding and the accumulation of particles in healthy tissues [64]. In general, active delivery limits the contact of MNPs with monocytes and macrophages, minimizes the delay and absorption of MNP by the reticuloendothelial system. Magnetic resonance imaging demonstrated that aptamer-functionalized arabinogalactan-coated superparamagnetic magnetite nanoparticles in healthy mice after intravenous injection were detected in the intestines 1.5 h after injection, where it might have come from the liver [63]. Blood serum biochemical parameters such as cholesterol, serum alanine amino-transferase, alkaline phosphatase and bilirubin did not change after three injections of aptamer-functionalized gold-coated MNPs to healthy male and female mice. According to the total protein concentration, the inflammation and hydration status of the animals treated with nanoparticles was not significantly different in mice treated with aptamer-coated MNPs and phosphate buffer saline as a control. Aptamer functionalization of gold-coated MNPs was safe and did not cause hepatotoxic effects [66]. Thus, the targeted delivery of magnetic particles using aptamers reduces their potential toxicity and increases biological effectiveness [63,66].

It has been experimentally shown that about 40% of MNPs are removed from the systemic blood circulation in the first 24 h with the urine. The distribution of MNPs in organs and tissues directly depends on the size of the particles. MNPs larger than 100 nm predominantly accumulate in the liver and spleen, MNPs of 50 nm in size penetrate BBB in vivo and accumulate in the brain [67,68], localizing in the neurons, astrocytes and macrophages [69,70]. However, most MNPs accumulate in the liver, spleen, kidneys, heart, and tumors, and in the lungs and testicles [60,67] (Figure 3). The accumulation of MNPs in tissues occurs not only by itself, but also as a result of their absorption by macrophages, with which they enter the liver, spleen, lungs and bone marrow [71]. ZnO MNPs, regardless of particle size, accumulate in the liver, lungs, and kidneys [72]. Aptamer-functionalized arabinogalactan-coated MNPs were removed with faeces and did not accumulate in organs except the intestines [66]. According to electron microscopy data, gold coated aptamer-functionalized MNPs did not accumulate in organs [63].

The typical final biodistribution of iron oxide nanoparticles without ligand functionalization is as follows: 80%–90% are in the liver, 5%–8%-spleen and 1%–2%-bone marrow [73]. All these organs contain large amounts of macrophages, which are part of the reticuloendothelial system (RES). Long-term in vivo studies in a mouse model have shown that nanoparticles coated with DMSA accumulate in the spleen, liver and lungs for a long time (at least up to 3 months) without any significant signs of toxicity. During this time, the nanoparticles undergo a biotransformation process by reducing the size or aggregation of the particles, or both. Low doses of magnetic nanoparticles are rapidly metabolized [74].

Aptamers significantly change the distribution of MNPs, because they are delivered by aptamers only to their molecular targets. Some data on the biodistribution of MNPs in comparison with the MNPs functionalized with aptamers are presented in Table 1.

## 4. Biodegradation and Excretion of Nanoparticles

MNPs are absorbed by cells, mainly through endocytosis. IONP-containing endosomes enter the lysosomes, where they are destroyed by a variety of hydrolytic enzymes, such as lysosomal cathepsin L [89]. MNPs functionalized with aptamers undergo biodegradation in the same way as unfunctionalized magnetic nanoparticles. Aptamers can be hydrolyzed under the influence of blood plasma nucleases or lysosome nucleases. In the acidic environment of lysosomes, iron is released from IONP and reduced [90]. Divalent iron ions are transported to the cytosol [91]. In the cytosol, iron ions bind to various proteins involved in the oxidation, storage and transport of iron [92]. At least five proteins are involved in the metabolism of iron in the cytosol, the expression of which increases in the presence of MNPs: L-ferritin (light chain of ferritin involved in iron binding and nucleation), H-ferritin (heavy chain of ferritin, with ferroxidase activity), ferroportin (iron export), divalent metal transporter DMT1 (transport of Fe (II) across cell membranes), and transferrin receptor TfR1 (iron import) [91]. The overexpression of genes encoding ferritin L and ferroportin occurs by the 3rd day and progressively increases in accordance with the rate of degradation of the MNP, without changing the expression of the ferritin H gene and the DMT1 gene [19]. Iron can be used in the cytosol to synthesize iron-containing proteins, in particular, heme in the mitochondria [93], and is exported from the cell by a transmembrane transporter (ferroportin). MNP causes an increase in the expression of ferritin and iron transport in macrophages, as well as an increase in the level of ferritin in endothelial cells [94] (Figure 4). Outside the cell, iron is oxidized and transported into the blood in complex with transferrin to those organs where the cells have a transferrin receptor on the cell membrane: hepatocytes in the liver, erythroblasts in the bone marrow, and red blood cells [95]. In the kidneys, there is a constant filtration of iron and its subsequent reabsorption in the loops of Henle. MNPs are rapidly absorbed by endothelial cells and macrophages of the liver, spleen, and atherosclerotic lesions after intravenous injection [96]. Most of the superparamagnetic iron of MNP decomposes within 30 days. Iron loading using MNPs induces a differential reaction of the mechanisms of cellular iron homeostasis with an increase in ferritin and iron transport proteins in macrophages and an increase in ferritin in the endothelial system.

The mass biodegradation of magnetic nanoparticles occurs within a month due to binding to ferritin and does not affect cellular iron homeostasis [74,75,97] (Figure 5). The size of the MNPs does not affect their biodegradation. Degradation occurs according to the “all or nothing” mechanism: nanoparticles either completely dissolve or remain intact. The structural and magnetic properties of the remaining nanoparticles after the decomposition process differ significantly from the properties of the initial suspension. The degradation kinetics are affected by pH, coating, and the average particle size of nanoparticles [98].

## 5. Conclusions

Magnetic nanoparticles are one of the most promising tools for nanomedicine, which are used as effective diagnostic and therapeutic means; MNPs are potentially toxic, since they accumulate in the liver, spleen, kidneys and other tissues, and also pass through the BBB; Aptamers increase the biocompatibility of MNPs and reduce their toxicity, since they are biocompatible, have a high affinity, and are highly specific, non-immunogenic and easily biodegradable.

## Figures and Tables

**Figure 1 biomedicines-08-00059-f001:**
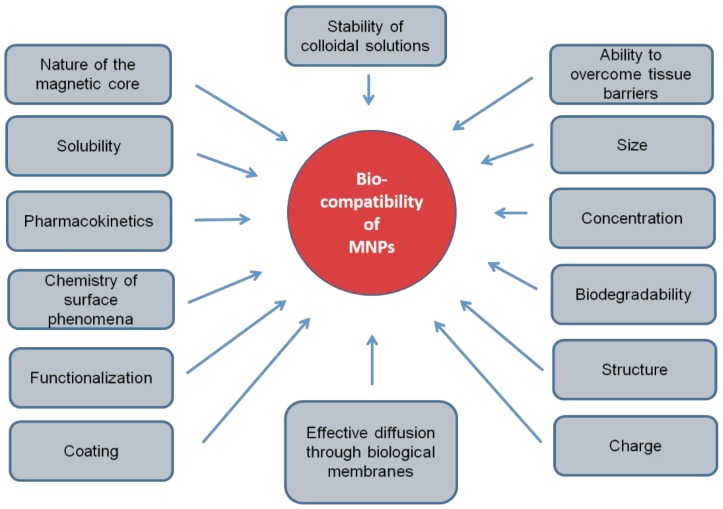
Physical and chemical factors of magnetic nanoparticles, determining their biocompatibility.

**Figure 2 biomedicines-08-00059-f002:**
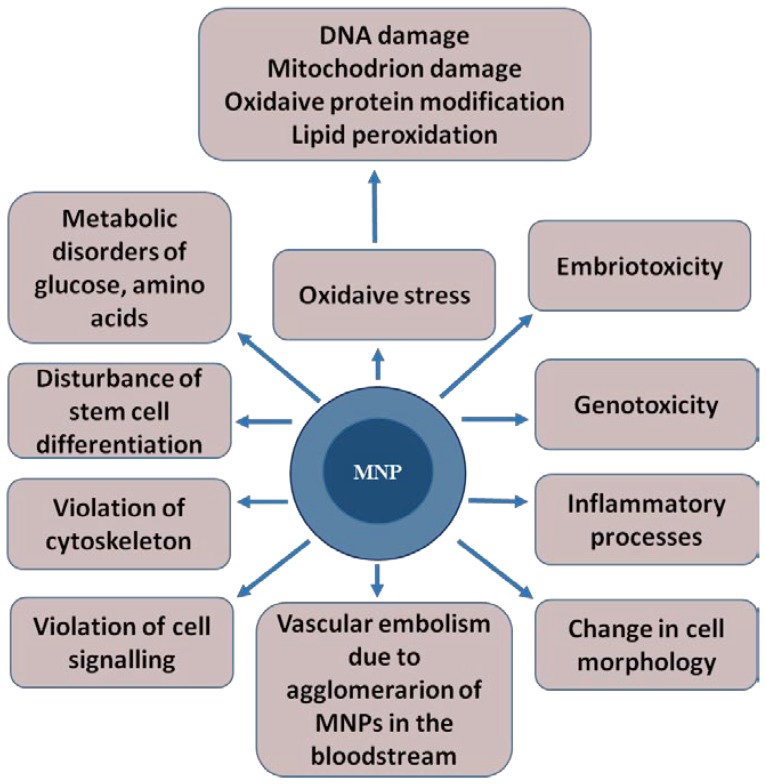
The toxic effects that magnetic nanoparticles can cause in a living organism. The magnetic core of the nanoparticle is shown in dark blue, and shell is represented in light blue.

**Figure 3 biomedicines-08-00059-f003:**
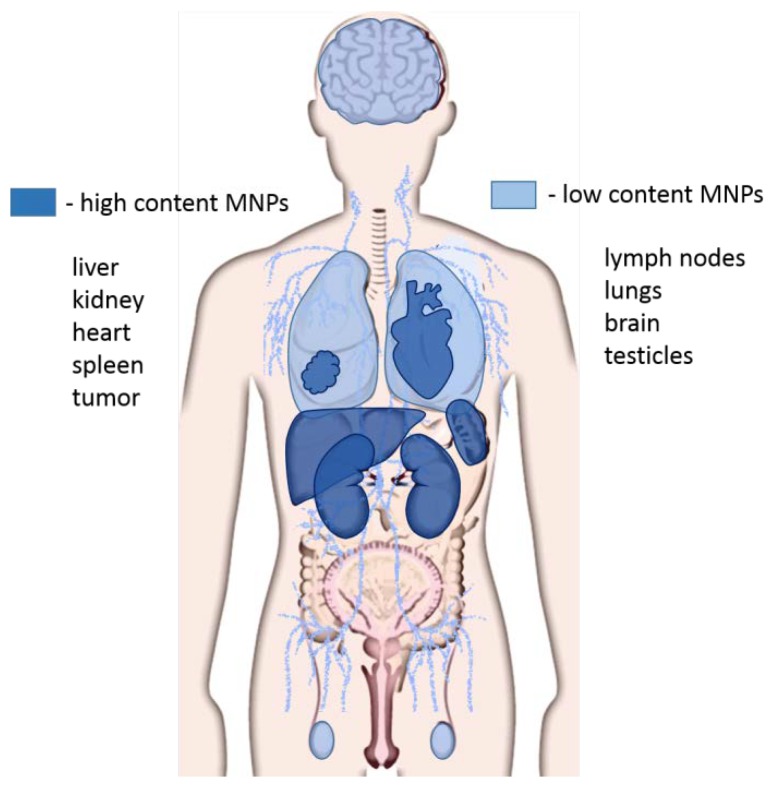
Distribution of magnetic nanoparticles in organs and tissues. The high concentration of magnetic nanoparticles (MNPs) accumulates in the liver, kidneys, spleen, heart, and tumors. Lower concentrations of MNPs are observed in the brain, lungs, lymph nodes, and testes.

**Figure 4 biomedicines-08-00059-f004:**
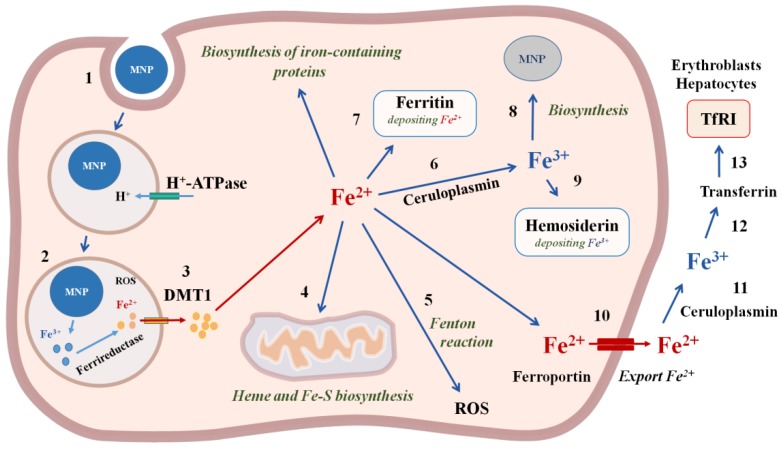
The mechanisms of iron magnetic nanoparticles incorporation and biodegradation in the cell. 1—MNP incorporation using endocytosis; 2—degradation of MNP and reduction in iron with ferrireductase; 3—Fe^2+^ transfer through the membrane with divalent metal transporter DMT1; 4—transport of Fe^2+^ into mitochondria for heme and Fe-S biosynthesis; 5—ROS formation in the Fenton reaction; 6—oxidation of Fe^2+^ with ceruloplasmin to Fe^3+^; 7—deposition of Fe^2+^ with ferritin; 8—biosinthesis of cellular MNPs; 9—deposition of Fe^3+^ with hemosiderin; 10—export of Fe^2+^ from the cell with ferroportin; 11—oxidation of Fe^2+^ with ceruloplasmin to Fe^3+^; 12—binding to transferrin; 13—Transferrin entry into the cells, which contain Transferrin receptors TfR1on the cell membranes, in particular, erythroblasts and hepatocytes.

**Figure 5 biomedicines-08-00059-f005:**
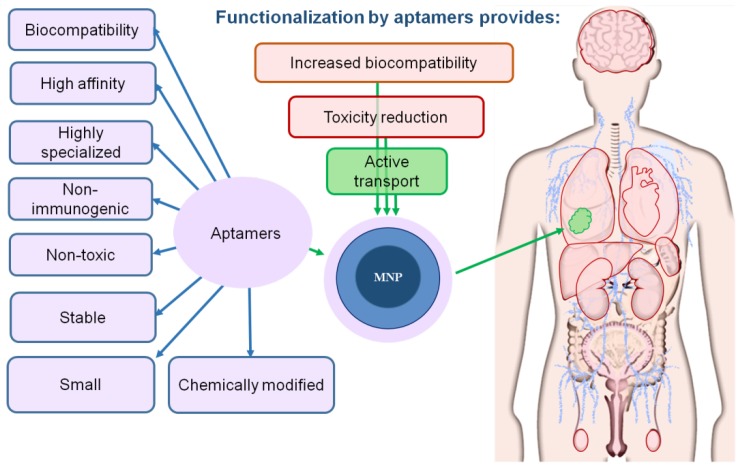
Aptamers increase the biocompatibility of magnetic nanoparticles. The magnetic core of the nanoparticle is shown in dark blue; the shell is represented in light blue; aptamer coating is light violet. The tumor is depicted in green.

**Table 1 biomedicines-08-00059-t001:** The distribution of MNPs in organs and tissues, depending on their size.

Particle Size	Target	Accumulation	Excretion	References
<10 nm		Lymph nodes	Urine, feces	[75]
10 nm		Liver, kidney, lung, spleen		[76]
5–20 nm		Lymph nodes	Urine, feces	[77]
>40 nm Large superparamagnetic iron oxide nanoparticles		Liver (80%), spleen (15%)	Liver (80%), spleen (15%)	[78]
<50 nm		Macrophages, RES		[79]
Dendrimers G4 @ IONP		Kidneys, liver, lungs, tumor		[33]
50 nm Ferromagnetic particles, coated with SiO2		Liver, spleen, lungs, testicles, kidneys, heart, brain		[60]
200 nm Ferromagnetic particles		Opsonization	Liver, spleen, lung, bone marrow macrophages	[80]
Non-biodegradable inorganic and polymeric micrometric spheres		Opsonization	Liver, spleen, lung, bone marrow macrophages	[81]
A cobalt-ferrite nanoparticle-AS1411 aptamer, 50 nm	Nucleolin	Breast tumor		[82]
A cobalt-ferrite nanoparticle–uMUC-1 aptamer, 50 nm	Underglycosylated mucin-1 (uMUC-1)	Tumor cells		[83]
Fe3O4@Au-AS1411 aptamer	Nucleolin	Breast tumor		[84]
Aptαvβ3-MNPs	αvβ3 Integrin	Tumor cells		[85]
Au @ SPION-MUC-1 aptamer	Underglycosylated mucin-1 (uMUC-1)	Colon tumor		[64]
Apt-MNPs	VEGFR2	Glioblastoma		[86]
USPIO-VEGF165-aptamer	VEGF165	Tumor		[87]
USPIO-AP613-1	glypican-3 (GPC3)	Hepatocellular carcinoma		[88]
AS-14-GMNPs	fibronectin	Mice Ehlich’s carcinoma		[63]
AS-FrFeAG	Fibronectin for AS-14, Hsc70 for As-42	Mice Ehlich’s carcinoma		[66]

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
