# Peer review of "Aptamers Increase Biocompatibility and Reduce the Toxicity of Magnetic Nanoparticles Used in Biomedicine"

_biomedicines, 2020, doi:10.3390/biomedicines8030059_

Round 1

Reviewer 1 Report

For the most part, this is an exceptionally well-written manuscript, particularly for a first submission. However, rarely have I enjoyed reading a manuscript this good only to be very disappointed that the main topic (according to the title) was only described over in a shockingly low number of text lines (lines 299-307) within about 10 pages of text (not including references). The authors need to describe the aptamer-MNP studies in more detail rather than providing just a summary statement for them. For example, was the half-life change 2-fold or 100-fold for aptamer coated MNPs? Was the toxicity 50 times less or only 2 times less? This review should go into much more detail so that the reader will know just how valuable adding an aptamer can be. In other words, can you quantify the improvement for the reader of this article?

An amazingly short list of other corrections follows:

Line 12: This small -> These small 

Table 2: "sbone" -> "bone" in bottom two rows

Lines 173-175 - remove this text entirely

Lines 248-251: Rewrite this sentence(s): "In the BV2..."

Line 281: that are affinity -> that have affinity

Line 301: (Fig.) -> (Fig. 5)?

Author Response

We thank you for the valuable suggestions and recommendations. We included into the text more detailed information about how aptamers enhance biocompatibility and reduce toxicity. We have added information about excretion routes. Unfortunately there is a limited number of studies on this topic. Even this is enough to demonstrate the ease and effectiveness of using aptamers as a ligands for targeting which also reduce toxicity of MNPs. We hope that this review will help those readers who are looking for the best ligand for targeting to make a good choice and use aptamers in their studies. The properties of aptamers that increase the biocompatibility of magnetic nanoparticles are discussed in this review.

  1. The text of the article has been reduced and substantially reformatted, the structure of the article has been changed, the order of presentation of the drawings has been changed. The part of the text that is not directly related to the aptamers has been deleted. The properties, biocompatibility and toxicity of MNP and aptamer-MNP were compared. Unfortunately, there is limited number of literature that compare the toxicity of MNPs and aptamer functionalized MNP. Basically, a comparison of MNPs and aptamer-MNP is based on their effectiveness. The article presents studies of the manifestations of the toxicity of MNPs and the reduction of their toxicity using functionalization of aptamers.
  2. The Review summarizes information from various sources on the biodegradation of iron-based MNPs. It is indicated that aptamers do not change the half-life, since they enter the cell in the same way as MNPs by endocytosis and the first stages of biodegradation occur in lysosomes. But if without aptamers biodegradation occurs in many organs and tissues, see the table, then coated with aptamers MNPs biodegrade only in target cells and, thus, reduce the toxicity of MNPs at the body level.
  3. The toxicity of MNPs is determined by the cell survival parameters in cell cultures; it should be at least 80%. MNPs are toxic at a concentration of 45-60 μg / ml, while aptamer-MNP do not show toxicity at higher concentrations (100-300 μg / ml and higher).
  4. Table 1 of the article has been deleted because there is no comparison between MNP and aptamer-MNP. Table 2 supplemented by aptamer-MNP data.

Reviewer 2 Report

The manuscript "Aptamers increase biocompatibility and reduce the toxicity of magnetic nanoparticles used in biomedicine" is written by Zamay et al. It seems that the content has no obvious relationship with the biocompatibility of aptamer-modified magnetic nanoparticles (MBs) except in the last part “Increase of biocompatibility of magnetic nanoparticles using aptamers”. Most characteristics of general MBs such as biodegradation and excretion were discussed but the information of aptamer-modified MBs was limited in the manuscripts. Thus, the manuscript must be substantially revised before resubmitting (including professional language editing). As it stands, it does not easily deliver its message : "aptamers increase biocompatibility and reduce the toxicity of magnetic nanoparticles" to the readers. In particular, you need to significantly shorten the manuscript in order to improve its comprehensibility. Further, you should focus solely on the biocompatibility and the toxicity of aptamer-modified MPs.

In addition, many relevant reviews of aptamer-modified MPs in biomedicine have been published, and thus what is the purpose of your review and what is the difference between yours and other reviews? The authors should clearly explain this part in the manuscript.

Author Response

We thank you for the valuable suggestions and recommendations. The text of the article is abridged and substantially reformatted. The structure of the article is changed, the order of presentation of the drawings is changed. The part of the text that is not directly related to aptamer-MNP has been removed. The properties, biocompatibility and toxicity of MNP and aptamer-MNP are given in comparison. Unfortunately, there is little data comparing the toxicity of MNPs and aptamer-MNP. Basically, a comparison of MNPs and aptamer-MNP is based on their effectiveness.

We included into the text more detailed information about how aptamers enhance biocompatibility and reduce toxicity. We have added information about excretion routes. Unfortunately there is a limited number of studies on this topic. No articles have been found comparing the biocompatibility and biodegradation of MNPs and aptamer-MNP. The presented review, first of all, aims to show the possibility of removing the restrictions on the use of MNPs that arise when used in biomedicine by introducing the procedure for their functionalization with aptamers, which have advantages over other types of ligands and reduce their toxicity several times. Even this is enough to demonstrate the ease and effectiveness of using aptamers as a ligands for targeting which also reduce toxicity of MNPs. We hope that this review will help those readers who are looking for the best ligand for targeting to make a good choice and use aptamers in their studies. The properties of aptamers that increase the biocompatibility of magnetic nanoparticles are discussed in this review.

Tab. 1 of the article has been deleted because there is no comparison between MNP and aptamer-MNP. Table 2 is supplemented by aptamer-MNP data.

Round 2

Reviewer 1 Report

This revised manuscript is much more focused than the original manuscript. Several areas need the addition of references, such as lines 172-179, where several claims are made. Two or more references should also be added in lines 241-244 [perhaps the same references used in the next paragraph?] 

Author Response

We thank reviewer for the edits. Everything has been corrected. 

Reviewer 2 Report

I am very happy that revised manuscript has been improved significantly in response to the reviewer comments, although I still find that English could be improved in the entire manuscript. For example,

Line 179   Magnetic resonance imaging demonstrate that…

Line 235,243,244,250,and 255 The word “f” of ferritin should be lowercase.

Author Response

We thank Reviewer for the edits. English language has been corrected.